# The Burden of Epstein–Barr Virus (EBV) and Its Determinants among Adult HIV-Positive Individuals in Ethiopia

**DOI:** 10.3390/v15081743

**Published:** 2023-08-15

**Authors:** Kidist Zealiyas, Seifegebriel Teshome, Nega Berhe, Wondwossen Amogne, Aklilu Feleke Haile, Ebba Abate, Getnet Yimer, Christoph Weigel, Elshafa Hassan Ahmed, Tamrat Abebe, Robert Baiocchi

**Affiliations:** 1Aklilu Lemma Institute of Pathobiology, Addis Ababa University, Addis Ababa 1176, Ethiopia; kzealiyas@gmail.com (K.Z.); nega.berhe.belay@gmail.com (N.B.); aklilu.feleke@aau.edu.et (A.F.H.); 2Ethiopian Public Health Institute, Addis Ababa 1242, Ethiopia; 3Department of Microbiology, Immunology and Parasitology, Addis Ababa University, Addis Ababa 9086, Ethiopia; seifegebriel.teshome@aau.edu.et (S.T.); tamrat.abebe@aau.edu.et (T.A.); 4Department of Internal Medicine, College of Health Sciences, Addis Ababa University, Addis Ababa 9086, Ethiopia; wonamogne@yahoo.com; 5Global One Health Initiative, Addis Ababa 1000, Ethiopia; waktola.2@osu.edu; 6Centre for Innovative Drug Development and Therapeutic Trials for Africa (CDT-Africa), College of Health Sciences, Addis Ababa University, Addis Ababa 9086, Ethiopia; getnetyimer@yahoo.com; 7Center for Global Genomics and Health Equity, Department of Genetics, Perelman School of Medicine, University of Pennsylvania, Philadelphia, PA 19104, USA; 8Comprehensive Cancer Center, The James Cancer Hospital and Solove Research Institute, The Ohio State University, Columbus, OH 43210, USA; christoph.weigel@osumc.edu; 9Division of Hematology, Department of Internal Medicine, College of Medicine, The Ohio State University, Columbus, OH 43210, USA

**Keywords:** DNA, Epstein–Barr virus, HIV/AIDS, viral capsid antigen, Ethiopia

## Abstract

Epstein–Barr virus (EBV) is a well-known risk factor for the development of nasopharyngeal carcinoma, Hodgkin’s lymphoma (HL), and Non-Hodgkin’s lymphoma (NHL). People with HIV infection (PWH) are at increased risk for EBV-associated malignancies such as HL and NHL. Nevertheless, there are limited data on the burden of EBV among this population group in Ethiopia. Hence, this study aimed to determine the burden of EBV infection among adult HIV-positive individuals in Ethiopia and assess the determinants of EBV DNA positivity. We conducted a cross-sectional study at the Tikur Anbessa Specialised Hospital from March 2020 to March 2021. Two hundred and sixty individuals were enrolled in this study, including 179 HIV-positive and 81 HIV-negative individuals. A structured questionnaire was used to capture demographic and individual attributes. In addition, the clinical data of patients were also retrieved from clinical records. EBV viral capsid antigen (VCA) IgG antibody was measured by multiplex flow immunoassay, and EBV DNA levels were tested by quantitative real-time polymerase chain reaction (q-PCR) assays targeting the *EBNA-1* open reading frame (ORF). Descriptive statistics were conducted to assess each study variable. A multivariable logistic regression model was applied to evaluate the determinants of EBV infection. Statistical significance was determined at a *p*-value < 0.05. Two hundred and fifty-three (97.7%) study participants were seropositive for the EBV VCA IgG antibody. Disaggregated by HIV status, 99.4% of HIV-positive and 93.8% of HIV-negative participants were EBV seropositive. In this study, 49.7% of HIV-positive and 24.7% of HIV-negative individuals were EBV DNA positive. PWH had a higher risk of EBV DNA positivity at 3.05 times (AOR: 3.05, 95% CI: 1.40–6.67). Moreover, among PWH, those with an HIV viral load greater than 1000 RNA copies/mL (AOR = 5.81, 95% CI = 1.40, 24.13) had a higher likelihood of EBV DNA positivity. The prevalence of EBV among PWH was significantly higher than among HIV-negative individuals. Higher HIV viral loads in PWH were associated with an increased risk of EBV DNA positivity. Since the increases in the viral load of EBV DNA among PWH could be related to the risk of developing EBV-associated cancers, it is necessary for more research on the role of EBV in EBV-associated cancer in this population group to be carried out.

## 1. Introduction

Viral co-infections in people with HIV infection (PWH) is a major global public health concern [1]. Co-infections with viruses have an increased association with the development of cancer in PWH [2]. Moreover, the natural history of these viral infections may be accelerated in PWH [3]. Epstein–Barr virus (EBV), hepatitis B virus (HBV), hepatitis C virus (HCV), human papillomavirus (HPV), and Kaposi’s sarcoma-associated herpesvirus (KSHV) are among the viruses that can lead to cancer in this population group [4]. The EBV, Human Gamma Herpesvirus 4, is a widely spread human virus carried as a life-long asymptomatic infection by most immune-competent individuals. Over 90% of the population is seropositive for EBV globally [5]. After primary infection, EBV establishes persistence in memory B lymphocytes, which can be detected in lymphoid tissue and peripheral blood [6]. The natural history of EBV infection in HIV-positive individuals is complex and varies depending on the severity of the HIV infection [7,8]. Studies have shown that EBV infection is more likely to be symptomatic and lead to serious complications in people with advanced HIV disease, such as EBV-associated lymphoproliferative disorders (LPD) [9]. LPDs are caused by the uncontrolled growth of EBV-infected B cells in people with HIV (PWH) [10]. Since the majority of LPDs contain EBV DNA and express viral gene products, such as EBV-encoded small RNAs (*EBER*), EBV plays a role in the pathogenesis of malignancies, including non-Hodgkin lymphoma (NHL) and Hodgkin lymphoma (HL) in PWH [7,10,11].

More than 70% of the world’s PWH are located in sub-Saharan Africa (SSA), the region that has been worst hit by the AIDS epidemic [12]. Ethiopia is one of the countries in SSA that has been most severely affected by HIV infections, with about 720,000 PWH and 27,104 newly diagnosed cases by 2020 [13]. The overall prevalence of HIV in Ethiopia is 0.96%, while the prevalence in urban areas is 3% [14,15]. In Ethiopia, cancer has become the second leading cause of death in the adult population [16].

Combination antiretroviral therapy (cART) has greatly modified the natural course of HIV infection, resulting in a decreased HIV RNA plasma viral load, increased absolute CD4 T cell count, and decreased HIV-associated opportunistic infections, indicating restoration of immune function [17]. However, the impact of cART seems less favorable regarding EBV-related malignancies than on other AIDS-defining cancers (ADCs), including EBV-positive Diffuse Large B Cell Lymphoma (DLBCL) and HL. NHL was the most common ADC during the cART era [18]. Discordant responses to cART with persisting HIV plasma viral load and chronic immune activation may contribute to sustaining B-cell activation and increasing EBV DNA levels [19].

EBV is a major hurdle in managing HIV/AIDS-infected individuals, since it is one of the most frequent causes of morbidity and mortality in this population [20]. In PWH, the early detection of EBV-related disease and intervention helped delay HIV/AIDS disease progression. However, minimal information is available on the prevalence of EBV and its determinants among PWH in Ethiopia. This study was therefore designed to determine the magnitude of EBV DNA positivity among adult PWH receiving cART at Tikur Anbessa Specialized Hospital Addis Ababa, Ethiopia.

## 2. Materials and Methods

### 2.1. Study Design, Time Period, and Setting

A cross-sectional study was conducted from March 2020 to March 2021 in Addis Ababa, Ethiopia, at the Tikur Anbessa Specialised Hospital (TASH). The hospital is a tertiary care teaching hospital of the Addis Ababa University located in Addis Ababa, the capital city of Ethiopia. The hospital offers inpatient, outpatient, and emergency services in twenty specialized clinics and units. The hospital has a bed capacity of 800, and more than 500,000 patients are treated as outpatients and inpatients annually.

### 2.2. Source and Study Population

The source population comprises adult PWH in Addis Ababa. The study population includes adult HIV-positive individuals on cART and outpatients in TASH who fulfilled all the inclusion criteria.

### 2.3. Inclusion and Exclusion Criteria

To be enrolled in the study, participants met the following inclusion criteria: adult HIV-positive individuals (18 years of age and older), who were willing to undergo testing for HIV (for the control group).

### 2.4. Study Variables

The dependent variable for this analysis was EBV DNA status among HIV-positive individuals in Addis Ababa, Ethiopia. The independent variables were socio-demographic, such as sex, age, place of residence, marital status, educational level, source of income, and average monthly payment, administered at the time of care, and clinical data (most recent HIV viral load, most recent CD4 cell count, ART, and regimen type) on patients obtained from clinical records.

### 2.5. Sample Collection

About 40 mL of venous blood was drawn from each participant using the acid citrate dextrose (ACD) anticoagulation-sterilized tubes. The sample was used for peripheral blood mononuclear cells (PBMC) and plasma isolation. A serum separator tube was used for serum collection. Blood samples were handled and stored following standard safety procedures and guidelines. All laboratory analysis was performed at the Ohio State University Comprehensive Cancer Center in the United States of America (Columbus, OH, USA).

### 2.6. Sample Preparation

Ficoll-Paque PLUS (Global Life Sciences Solutions USA LLC, Carlsbad, CA, USA) was used to isolate lymphocytes from blood samples (*n* = 260) collected in ACD tubes per the manufacturer’s instructions. Briefly, 1× phosphate-buffered saline (PBS) balanced salt solution (Life Technologies, Carlsbad, CA USA) was used for 1:1 dilution of the blood samples, and the diluted blood was layered over the Ficoll-Paque PLUS solution. After centrifugation at 350 rcf for 30 min at 20 °C, PBMCs were collected in a separate tube. Finally, PBMCs were subjected to short washing steps using a balanced salt solution to remove platelets, Ficoll-Paque PLUS, and remaining plasma. Isolated PBMCs were stored at −80 °C in 10% DMSO until DNA extraction was performed.

### 2.7. HIV Serology Test

For the control group, HIV-1/2 infection was screened by immune chromatographic or immune filtration format via rapid diagnostic test (RDT) as per the national testing algorithm. Counseling was provided to each study participant who consented to be enrolled in this study. Following the counseling, RDT was performed for HIV testing: STAT PACK^TM^ (screening test), ABON^TM^ (confirmatory test), and SDBIOLINE^TM^ (tie-breaker test). The results of the RDTs were disclosed to the participants, and as part of the routine procedure, all clients found to be HIV-positive per the national algorithm were referred to HIV care and treatment services (Figure 1).

### 2.8. EBV Serology Test

Quantitative EBV-specific serology was performed on serum for all samples. EBV viral capsid antigen (VCA) IgG antibodies were determined by a multiplex flow immunoassay (MFI) using the BioPlex 2200 EBV IgG kits on a BioPlex 2200 Analyzer (Bio-Rad, Hercules, CA USA), according to the manufacturer’s instructions. Results were then classified, according to their antibody index values, as unfavorable (≤0.8), equivocal (0.9 to 1.0), or positive (≥1.1).

### 2.9. DNA Extraction

DNA was isolated from 5 × 10^6^ PBMCs in 200 μL using the QIAamp DNA mini kit (QIAGEN, Germantown, MD, USA), as per the manufacturer’s instructions. The concentration and purity of all extracted DNA samples were measured using a Qubit 3.0 fluorometer (Life Technologies, Waltham, MA, USA) and stored at −80 °C until used for quantitative real-time polymerase chain reaction (qPCR).

### 2.10. Real-Time Quantification PCR from Genomic DNA for EBV Load/Copy Number Determination

EBV was detected using the *EBNA1* gene and performed with 10 ng of sample DNA as the initial concentration using the ViiA7 Real-time qPCR machine (Applied Biosystems, Waltham, MA USA). EBV DNA was quantified with primers specific to the EBV *EBNA1* locus (forward: TCATCATCATCCGGGTCTCC, reverse: CCTACAGGGTGGAAAAATGGC), and signals were normalized to host genome DNA using primers specific for human *ACTB* (forward: CAGGCAGCTCGTAGCTCTTC, reverse: TCGTGCGTGACATTAAGGAG) [22]. With a total reaction volume of 10 µL, the reaction was carried out using 5 µL of 2× Fast SYBR Green Master Mix (Applied Biosystems, Waltham, MA USA), 0.25 µL of forward (10 µM) and 0.25 µL of reverse (10 µM) primers, 2.5 µL of PCR-grade water, and 2 µL of 5 ng/µL DNA concentration. The qPCR consisted of 40 cycles (95 °C for 1 s, 60 °C for 20 s, and 70 °C for 30 s). The positive control was the Raji cell line and the negative control was the K-562 cell line, which were included in each reaction. To calculate the EBV viral load, twelve standards of the known DNA copy number for the *EBNA1* gene and *ACTB* loci at various concentrations (with 2-fold dilutions) were used to calculate EBV copies per ml. Each sample and the corresponding standard were run in triplicates on 384-well PCR plates. The *EBNA1* gene’s CT value was calculated and converted into EBV copies/mL to obtain the EBV copy number. Genome copies per cellular genome elevated by more than two-fold above the negative control were considered EBV DNA positive. We used a no-template control (NTC) sample to detect qPCR reaction contamination in each run. All samples and controls were run in triplicate. The detection limit was determined using a purified PCR product in a dilution series and then determining the limit of detection with EBV-negative control DNAs.

### 2.11. Data Collection and Quality Assurance

The data were collected using a standardized data collection form. The form was pre-tested on 5% of the sample that was not used in the study. Under the direction of the principal investigator, a study nurse collected socio-demographics using a structured questionnaire and clinical data from a patient’s medical chart. Data collectors received training before data collection to help them incorporate the objective of the study, the method used, and the contents of the data extraction form.

### 2.12. Statistical Analysis

Data were entered into Epi Info 7, and statistical analyses were performed using IBM SPSS Statistics for Windows v. 26.0 (IBM).

Descriptive statistics were used to summarize the study variables. All variables were summarized using frequencies and proportions. The categorical data were compared using the chi-square test. Logistic regression models, such as bivariable and multivariable logistic regressions, were employed to determine the associations between variables. We assessed the independence, linearity, and normality tests before considering the multivariable regression analysis to ensure that our model was normally and randomly distributed. The odds ratio was determined along with the 95% CI, and those variables with a *p*-value <0.2 in the bivariable analysis were subjected to multivariable analysis to identify the independently associated variables. To account for the differences between the control and study groups, we used confounders related to demographic characteristics during the multivariable-adjusted odds ratio analysis. A *p*-value < 0.05 in the multivariable analysis was considered a statistically significant association. 

## 3. Results

### 3.1. Socio-Demographic Characteristics

A total of 260 study participants, including 179 PWH on cART and 81 HIV-negative individuals, were included in the study. HIV infection was screened via the immune chromatographic or immune filtration format based on the national testing algorithm (Figure 1). Most participants (163, 62.7%) were female, and nearly one-half (127, 48.8%) were in the 18–38-year-old age group. More than one-third (105, 40.4%) of participants were married. About two-thirds (170, 65.4%) had primary and secondary school education, one-third (88, 33.8%) were government employees, and more than half (140, 53.8%) had a monthly income of ETB 3000 (USD 55.24) or less (Table 1).

### 3.2. Detection of EBV Antibody

Of all the study participants, 253 (97.7%) were seropositive for the EBV VCA IgG antibody. Among the 179 HIV-positive individuals, 178 (99.4%) were seropositive, and 75 (93.8%) of the HIV-negative individuals were seropositive. There was a significant difference in seroprevalence between HIV-positive and -negative individuals (99.4% vs. 93.8%; *p*-value, 0.018) (Figure 2). EBV seroprevalence increased with age. Among the age group of 18–27 and 48 and above, EBV seropositivity was 93.8% and 100%, respectively (Figure 3).

### 3.3. EBV DNA Positivity by Demographic and Clinical Characteristics among HIV-Positive Individuals

In this study, EBV DNA positivity was 49.7% (*n* = 89) among HIV-positive individuals and 24.7% (*n* = 20) among HIV-negative participants (Figure 4). Among HIV seropositive individuals, EBV DNA positivity correlated with some socio-demographic and clinical characteristics of the participants, as shown in Table 2. More specifically, participants of ≥35 years of age and with high (>1000 RNA copies/mL) HIV viral load demonstrated significant associations with EBV DNA positivity.

### 3.4. Associated Factors of EBV DNA Positivity among HIV-Positive and HIV-Negative Individuals

In the bivariable regression analysis, age group, marital status, educational status, average monthly income, and HIV status were risk factors for EBV DNA positivity at a *p*-value of less than 0.2. When all these variables were subjected to multivariable regression analysis, only being HIV-positive was found to be statistically significant. Accordingly, the odds of being EBV positive among HIV-positive individuals were three times as high as [AOR = 3.05; 95% CI: 1.40–6.67] those among HIV-negative individuals (Table 3).

We then further analyzed the factors associated with EBV DNA positivity among PWH. Our results show that age group, marital status, source income, average monthly income, HIV viral load, and regimen type were identified in the bivariable regression analysis as risk variables for EBV DNA positivity at a *p*-value of less than 0.2. Only HIV viral loads with more than 1000 RNA copies per ml were shown to be statistically significant when all these factors were examined through multivariable regression analysis. Among PWH, those with an HIV viral load of greater than 1000 RNA copies/mL (AOR = 5.81, 95% CI = 1.40, 24.13) had a higher likelihood of EBV DNA positivity compared to those who had an HIV viral load of less than 1000 RNA copies/mL (Table 4).

## 4. Discussion

Our study demonstrated that the EBV VCA IgG antibody seroprevalence was higher in HIV-positive individuals. The chi-square test (*p* values = 0.018) and Fisher’s exact test (*p* values = 0.012) showed that the two groups are statistically different. However, the antibody level between the two groups is almost equal. This might be due to the small sample size of the control group, and the statistically significant differences might not have biological meaning.

EBV DNA positivity was significantly higher in HIV-positive individuals in Ethiopia. Being HIV-positive is a risk factor for EBV DNA positivity. Patients with an HIV viral load greater than 1000 RNA copies/mL had a higher likelihood of EBV DNA positivity. This is the first study to report these findings in Ethiopia, where HIV and cancer remain significant public health problems.

EBV-associated lymphomas are a considerable threat to individuals infected with HIV and constitute an ADCs [23]. With more than 90% of the adult human population infected with the oncogenic herpesvirus EBV [24], coinfection with this virus is common for PWH [25]. Accordingly, knowing the burden of productive EBV infection among PWH is essential for controlling morbidity and mortality and preventing complications [26].

The findings of this study revealed that the overall seroprevalence of the EBV VCA IgG antibody was significantly higher in HIV-positive individuals compared to HIV-negative individuals. Our results are consistent with other studies that have found a higher seroprevalence of EBV infection in HIV disease [27,28]. A study conducted in Iran found that the seroprevalence of EBV infection was 100% in HIV-positive individuals and 91.9% in HIV-negative individuals [26]. Similarly, a study conducted in South Africa found that the seroprevalence of EBV infection was 100% in HIV-positive individuals [29]. Other studies have found seroprevalence rates of 87.2% in Ghana, 95% in Iran, and 97.9% in Qatar among healthy blood donors [30,31,32]. This high rate of infection could be due to immunodeficiency.

In the current study, the EBV DNA positivity was comparable with other studies conducted in Africa (South Africa, Ghana, Uganda, and Malawi), which ranged from 36% to 45% [33,34,35,36]. When the EBV DNA positivity among HIV-positive individuals compared with HIV-negative individuals, the findings were consistent with studies conducted in China (56% vs. 26%) [37], the UK (42.1% vs. 16.6%) [38], and Brazil (48.3% vs. 51.7%) [39]. On the other hand, our findings were less than those reported in a study in the United States (81% vs. 16%) [8]. In contrast to our findings, EBV DNA positivity was 30% among PWH in India [11], 39.5% among healthy individuals in Japan [40], and 37.2% in Portugal [41]. A possible reason for differences in EBV DNA detection could be due to a lack of standardization testing. These may include variations in the degree of EBV DNA detection in different laboratories due to differences in diagnostic kits, equipment, procedures, sample types [42], immunity [10], and geographic factors [43].

Other studies reported a high prevalence of EBV DNA in PWH and showed that HIV viral load was a key factor for EBV reactivation [39,44]. The prevalence of EBV DNA was significantly correlated with a detectable HIV viral load in the plasma, as observed in other studies, including in PWH without lymphoproliferative diseases [44,45]. A study suggested an interaction between EBV and HIV in the regulation of replication [46]. However, the mechanism of this regulation was not fully explored. Our findings demonstrate a significant association between EBV positivity and increased HIV viral loads in PWH in Ethiopia. This analysis strengthens the implications of an HIV–EBV interaction.

The major finding of our analysis indicated that being HIV-positive was one of the risk factors for EBV DNA positivity compared to HIV-negative individuals. Cancer is now the second most significant cause of death for adults in Ethiopia [16,47]. The HIV status of people affected by cancer has not been discussed in the literature. It would be interesting to follow these EBV DNA-positive patients, evaluate the rate of lymphoma development, and design appropriate control measures. PWH have a higher likelihood of detectable EBV DNA, which may account for the high prevalence of EBV-associated lymphomas in this population. Thus, our findings confirm that HIV infection a risk factor for EBV serostatus and infectious processes (the presence of EBV DNA), consistent with the results of other studies [7,10,26,37].

## 5. Conclusions

In conclusion, this study described the burden of EBV and its determinants among adult HIV-positive individuals in Ethiopia. The EBV load among PWH was significantly higher than that of HIV-negative individuals. Higher HIV viral loads in PWH were associated with an increased risk of EBV DNA positivity. The increased EBV viral load among PWH could be related to a higher risk of developing EBV-associated cancers. Hence, there is a need to do more research on the role of EBV in EBV-associated cancer in this population group.

## Figures and Tables

**Figure 1 viruses-15-01743-f001:**
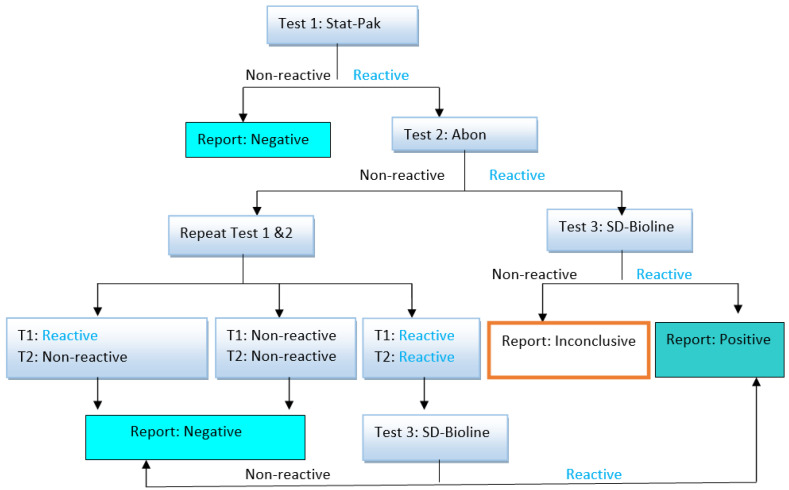
HIV diagnosis serial testing algorithm [21].

**Figure 2 viruses-15-01743-f002:**
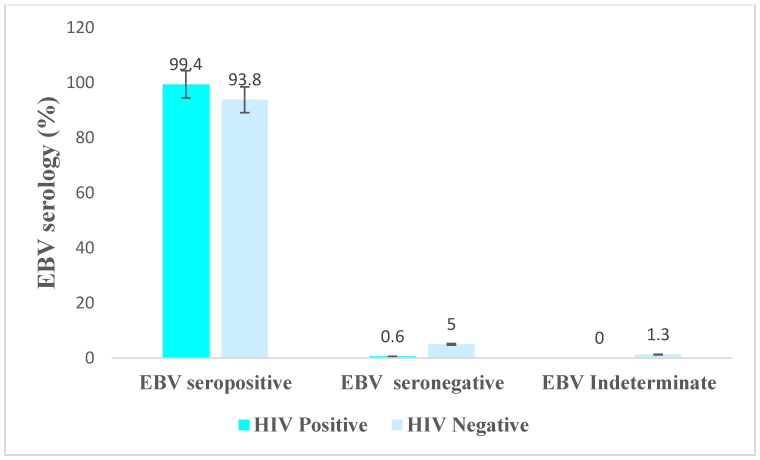
EBV serology with HIV status.

**Figure 3 viruses-15-01743-f003:**
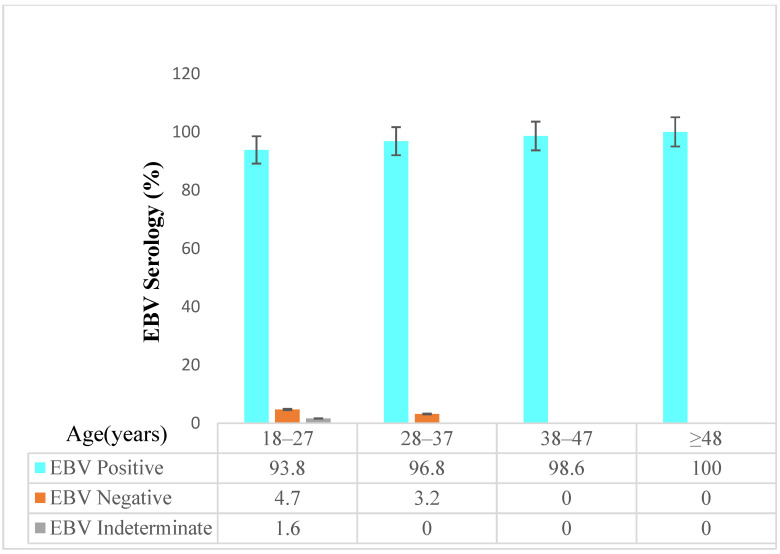
EBV serology with age.

**Figure 4 viruses-15-01743-f004:**
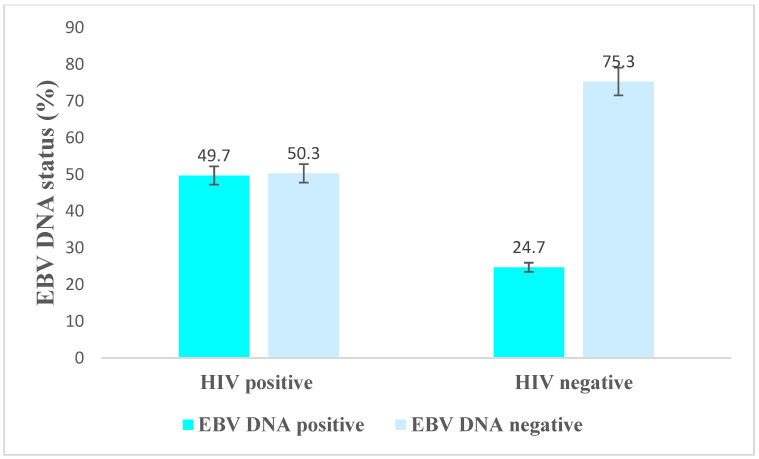
EBV DNA positivity among HIV-positive and -negative study participants.

**Table 1 viruses-15-01743-t001:** Socio-demographic characteristics of the study participants.

Characteristics		HIV Status				Total (260)	%
		Positive (*n* = 179)	%	Negative (*n* = 81)	%		
Sex	Male	66	36.9	31	38.3	97	37.3
	Female	113	63.1	50	61.7	163	62.7
Age	18–28	25	14.0	39	48.1	64	24.6
	29–38	38	21.2	25	30.9	63	24.2
	39–48	59	33.0	12	14.8	71	27.3
	>49	57	31.8	5	6.2	62	23.8
Marital status	Unmarried	51	28.5	46	56.8	97	37.3
	Married	74	41.3	31	38.3	105	40.4
	Divorced	23	12.8	4	4.9	27	10.4
	Widowed	31	17.3	0	0.0	31	11.9
Educational status	Illiterate	16	8.9	0	0.0	16	6.2
	Primary School	57	31.8	9	11.1	66	25.4
	Secondary School	76	42.5	28	34.6	104	40.0
	College diploma and above	30	16.8	44	54.3	74	28.5
Source of income	Government employee	26	14.5	62	76.5	88	33.8
	Stay-at-home spouse	27	15.1	0	0.0	27	10.4
	Private sector employee	53	29.6	0	0.0	53	20.4
	Self-employed	43	24.0	0	0.0	43	16.5
	Student	18	10.1	1	1.2	19	7.3
	Unemployed	12	6.7	18	22.2	30	11.5
Average monthly income in ETB	<1000	52	29.1	1	1.2	53	20.4
	1001–2000	37	20.7	13	16.0	50	19.2
	2001–3000	25	14.0	12	14.8	37	14.2
	3001–4000	15	8.4	26	32.1	41	15.8
	>4000	19	10.6	11	13.6	30	11.5
	NA	31	17.3	18	22.2	49	18.8

HIV: human immunodeficiency virus; ETB: Ethiopian birr; NA: not applicable.

**Table 2 viruses-15-01743-t002:** Epstein–Barr virus (EBV) DNA positivity among HIV-1 positive individuals by demographic and clinical characteristics.

Characteristics		EBV DNA Status				Total	%	*X* ^2^	*p*-Value
		Positive	%	Negative	%				
Sex	Male	30	45.5%	36	54.5%	66	37%	0.76	0.383
	Female	59	52.2%	54	47.8%	113	63%		
Age	<35	20	36.4%	35	63.6%	55	31%	0.57	0.017
	≥35	69	55.6%	55	44.4%	124	69%		
Marital status	Unmarried	23	45.1%	28	54.9%	51	28%	6.85	0.077
	Married	33	44.6%	41	55.4%	74	41%		
	Divorced	11	47.8%	12	52.2%	23	13%		
	Widowed	22	71.0%	9	29.0%	31	17%		
Educational status	Illiterate	9	56.3%	7	43.8%	16	9%	1.53	0.676
	Primary school	31	54.4%	26	45.6%	57	32%		
	Secondary school	34	44.7%	42	55.3%	76	42%		
	College diploma and above	15	50.0%	15	50.0%	30	17%		
Source of income	Government employee	15	57.7%	11	42.3%	26	15%	12.62	0.027
	Stay-at-home spouse	17	63.0%	10	37.0%	27	15%		
	Private sector employee	25	47.2%	28	52.8%	53	30%		
	Self-employed	5	27.8%	13	72.2%	18	10%		
	Student	2	16.7%	10	83.3%	12	7%		
	Unemployed	25	58.1%	18	41.9%	43	24%		
Average monthly income in ETB	<1000	29	55.8%	23	44.2%	52	29%	2.67	0.751
	1001–2000	20	54.1%	17	45.9%	37	21%		
	2001–3000	12	48.0%	13	52.0%	25	14%		
	3001–4000	7	46.7%	8	53.3%	15	8%		
	>4000	9	47.4%	10	52.6%	19	11%		
	NA	12	38.7%	19	61.3%	31	17%		
Most recent VL	<1000	74	46.5%	85	53.5%	159	89%	5.76	0.016
	>1000	15	75.0%	5	25.0%	20	11%		
Most recent CD4count	<200	14	51.9%	13	48.1%	27	16%	0.96	0.757
	≥200	70	46.7%	74	51.4%	144	84%		
ART	1st line	64	50.0%	64	50.0%	128	72%	2.57	0.277
	2nd line	19	43.2%	25	56.8%	44	25%		
	3rd line	4	80.0%	1	20.0%	5	3%		
Regimen type	ATV/r (PI based)	19	43.2%	25	56.8%	44	25%	4.70	0.195
	Darunavir (PI based)	4	80.0%	1	20.0%	5	3%		
	DTG (INSTI based)	62	51.7%	58	48.3%	120	68%		
	EFV/NVP (NNRTI based)	2	25.0%	6	75.0%	8	5%		

NA = Not applicable.

**Table 3 viruses-15-01743-t003:** Bi-variable and multivariable regression analysis of EBV DNA positivity among HIV-positive and HIV-negative individuals in Addis Ababa, Ethiopia (*n* = 260).

Characteristics		COR (95% CI)	AOR (95% CI)
Age	<35	1	1
	≥35	1.75 (1.06–2.89)	1.34 (0.65–2.72)
Marital status	Unmarried	1	1
	Married	1.04 (0.59–1.84)	0.78 (0.37–1.64)
	Divorced	1.16 (0.49–2.78)	0.77 (0.28–2.12)
	Widowed	4.14 (1.72–9.97)	2.71 (0.95–7.71)
Educational status	Illiterate	1.43 (4.83–4.25)	0.39 (0.11–1.50)
	Primary school	0.99 (0.51–1.92)	0.49 (0.20–1.19)
	Secondary school	0.54 (0.29–1.00)	0.32 (0.14–0.69)
	College diploma and above	1	1
Average monthly income	<1000	1.75 (0.80–3.84)	1.28 (0.503–3.24)
	1001–2000	0.967 (0.43–2.16)	1.03 (0.404–2.62)
	2001–3000	0.79 (0.33–1.90)	0.76 (0.285–2.03)
	3001–4000	0.41 (0.16–1.04)	0.49 (0.17–1.44)
	>4000	2.17 (0.86–5.40)	2.08 (0.72–6.01)
	NA	1	1
HIV status	Negative	1	1
	Positive	3.02 (1.68–5.41)	3.05 (1.4–6.67)

Crude odds ratio = COR; adjusted odds ratio = AOR, Not applicable = NA.

**Table 4 viruses-15-01743-t004:** Bi-variable and multivariable regression analysis of EBV DNA positivity among HIV-positive individuals in Addis Ababa, Ethiopia (*n* = 179).

Characteristics		COR (95% CI)	AOR (95% CI)
Age	<35	1	1
	≥35	2.20 (1.14–4.22)	2.16 (0.89–5.30)
Marital status	Unmarried	1	1
	Married	0.98 (0.48–2.01)	0.49 (0.172–1.40)
	Divorced	1.12 (0.42–3.00)	0.63 (1.90–2.11)
	Widowed	2.98 (1.15–7.71)	1.37 (0.40–4.71)
Source of income	Government employee	1	1
	Stay-at-home spouse	1.25 (0.41–3.75)	1.94 (0.504–7.50)
	Private sector employee	0.66(0.25–1.69)	0.92(3.15–2.72)
	Self-employed	0.28 (0.08–1.03)	0.35 (0.078–1.56)
	Student	0.147 (0.03–0.81)	0.154 (0.014–1.64)
	Unemployed	1.02 (0.38–2.73)	0.97 (0.27–3.50)
Average monthly income	<1000	2.0 (0.81–4.94)	1.31 (0.37–4.69)
	1001–2000	1.86 (0.71–4.91)	1.11 (0.28–4.47)
	2001–3000	1.46 (0.503–4.25)	0.94 (0.21–4.15)
	3001–4000	1.38 (0.40–4.81)	1.25 (0.21–7.48)
	>4000	1.43 (0.45–4.52)	1.11 (0.21–5.76)
	NA	1	1
Most recent VL	<1000	1	1
	>1000	3.45(1.20–9.94)	5.81 (1.4–24.13)
Regimen type	ATV/r based (PI based)	1	1
	Darunavir based (PI based)	2.30 (0.41–12.60)	2.4 (0.185–31.1)
	DTG based (INSTI based)	12.00 (0.80–180.9)	1.5 (0.66–3.38)
	EFV/NVP based (NNRTI based)	3.21 (0.62–16.53)	0.297 (0.039–2.28)

Crude odds ratio = COR; adjusted odds ratio = AOR, Not applicable = NA.

## Data Availability

The raw data can be obtained from the first author and corresponding author upon request.

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
