# Peer review of "The Burden of Epstein–Barr Virus (EBV) and Its Determinants among Adult HIV-Positive Individuals in Ethiopia"

_viruses, 2023, doi:10.3390/v15081743_

Round 1
Reviewer 1 Report
The manuscript by Zealiyas et al., describes an interesting topic; namely, the interplay between HIV infection and the status of Epstein-Barr virus (EBV). HIV infection is likely to increase the incidence of EBV-associated lymphoproliferative diseases especially in sub-Saharan Africa. Detection of EBV DNA in the plasma of HIV patients with NHL was identified as a poor prognostic factor. In this manuscript, Zealiyas and colleagues assessed the burden of EBV infection among HIV positive individuals in Ethiopia and identified statistically significant differences in EBV sero-positivity among HIV-infected patients relative to HIV-negative ones, 99.4% versus 93.8%, respectively. Additionally, the authors found that 49.7% of HIV positive individuals had plasma EBV DNA compared to 24.7% HIV negative individuals; interestingly, higher HIV viral loads (>1000 RNA copies/ml) correlated with detection of plasma EBV DNA. It remains to be established whether the increase in EBV load in PWH in Ethiopia correlates with a higher risk of developing EBV-associated cancers.
Comments:
1) While the difference in the seroprevalence of EBV VCA IgG antibody between HIV positive (99.4%) and HIV negative individuals (93.8%) was statistically significant, the two numbers are almost equal and are not biologically meaningful.
2) Figure 4, refers to EBV DNA negative HIV-/+ patients and assesses their percent of EBV DNA positivity. Its not clear how patients who are EBV DNA negative would still have EBV DNA up to 50.3% and 75.3%.
The manuscript in general is well written except for few sentences with stylistic issues, for example, lines 263-267.
Author Response
Response to Reviewer # 1 Comments:
- While the difference in the seroprevalence of EBV VCA IgG antibody between HIV positive (99.4%) and HIV negative individuals (93.8%) was statistically significant, the two numbers are almost equal and are not biologically meaningful.
Author Response:
We appreciate this point and have addressed this with new text added to the discussion (see red/italicized text below and in the manuscript: Lines 267-272).
Our study demonstrated that the EBV VCA IgG antibody seroprevalence was higher in HIV-positive individuals. The Chi-square test (p values = 0.018) and Fisher's exact test (p values = 0.012) showed that the two groups are statistically different. However, the antibody level between the two groups is almost equal. This might be due to the small sample size of the control group, and the statistically significant differences might not have biological meaning.
- Figure 4 , refers to EBV DNA negative HIV-/+ patients and assesses their percent of EBV DNA positivity. Its not clear how patients who are EBV DNA negative would still have EBV DNA up to 50.3% and 75.3%.
Author Response:
The EBV DNA positivity among HIV-positive and HIV-negative people is shown in Figure 4. EBV DNA positivity rates were 49.7% and 24.7%, respectively, for those with HIV and those HIV-negative. While 50.3% and 75.3% of HIV-positive and HIV-negative people had EBV DNA negative, respectively.
Comments on the Quality of English Language
- The manuscript in general is well written except for few sentences with stylistic issues, for example, lines 263-267.
Author Response:
Comment accepted and addressed. Please see red/italicized text below and in the manuscript: Lines 292-298.
Our results are consistent with other studies that have found a higher seroprevalence of EBV infection in HIV disease (1, 2). A study conducted in Iran found that the seroprevalence of EBV infection was 100% in HIV-positive individuals and 91.9% in HIV-negative individuals (3). Similarly, a study conducted in South Africa found that the seroprevalence of EBV infection was 100% in HIV-positive individuals (4). Other studies have found seroprevalence rates of 87.2% in Ghana, 95% in Iran, and 97.9% in Qatar among healthy blood donors (5-7).
Reviewer 2 Report
The paper by Zealiyas et al. describes high EBV seroprevalence and reactivation rate in a cohort of the Ethiopian HIV-positive patients when compared with healthy adult HIV-negative controls. Despite being studied in Ethiopia for the first time, high prevalence of EBV infection and elevated EBV DNA load in peripheral blood of HIV- positive individuals is well-known phenomenon. Hence the study originality and scientific contribution is low. Moreover, the study has several shortcomings which might have an impact on the results interpretation. Same statements in the text are incorrect or are wrong formulated. Therefore, I do not recommend this paper for the publication in the Viruses.
Main comments:
The control group is not matched with the study group of HIV-positive patients. It differs significantly in the age distribution and socio-economical state, both the factors playing important role in the prevalence of EBV infection.
In house PCR, that was used for quantitation of EBV DNA in blood samples, is not sufficiently described: performance characteristics (sensitivity, specificity, detection limit and quality control) are not specified. Why qualitative results only are presented? How many EBV DNA copies/ ml ( cut-off) were discriminative between positive and negative samples?
Other comments:
Introduction:
Nature history of EBV infection in HIV- infected individuals should be described in more detail to explain how it is accelerated.
Material and methods:
2.6. Sample preparation: In centrifugation give rcf value instead of rpm
2.9. DNA extraction: Give the sample amount in number of PBMC instead of μl of PBMC
Discussion:
Presence of EBV DNA in peripheral blood is a marker of reactivation, not co-infection. All EBV- seropositive PWH are co-infected. ( line 282).
Conclusions:
EBV load in PWH was significantly higher.
Abstract:
EBV is a risk factor for HL and NHL only in HIV- positive individuals. Rather than at higher risk of acquiring EBV infection PWH lose immune control of EBV latency.
Author Response
Response to Reviewer # 2 Comments:
Main comments:
- The control group is not matched with the study group of HIV-positive patients. It differs significantly in the age distribution and socio-economical state, both the factors playing important role in the prevalence of EBV infection.
Author Response:
This point is valid, and we incorporated text in the Material and Methods/statistical analysis section to clarify our confounder analysis approach (see red/italicized text below and in the manuscript: Lines 202-204).
To account for the difference between the control and study groups, we hold confounders related to demographic characteristics during multivariable-adjusted odds ratio analysis.
- In house PCR, that was used for quantitation of EBV DNA in blood samples, is not sufficiently described: performance characteristics (sensitivity, specificity, detection limit and quality control) are not specified. Why qualitative results only are presented? How many EBV DNA copies/ ml (cut-off) were discriminative between positive and negative samples?
Author Response:
We appreciate this comment and direct the reviewer to the end of the 2.10 section, where we describe the use of positive and negative control and twelve standards with known copy numbers for the two genes. To clarify, we have inserted additional text in the manuscript: Lines 179-184.
Our criteria for EBV DNA positivity were a 2-fold or greater elevation in genome copies per cellular genome compared to the negative control.
Other comments:
Introduction:
- Nature history of EBV infection in HIV- infected individuals should be described in more detail to explain how it is accelerated.
Author Response:
This point is valuable, and we have incorporated text to describe the nature history of EBV infection in HIV (see red/italicized text below and in the manuscript: Lines 62-70).
The natural history of EBV infection in HIV-positive individuals is complex and varies depending on the severity of the HIV infection (7, 8). Studies have shown that EBV infection is more likely to be symptomatic and lead to serious complications in people with advanced HIV, such as EBV-associated lymphoproliferative disorders (LPD) (9). LPDs are caused by the uncontrolled growth of EBV-infected B cells in people with HIV (PWH) (10). Since the majority of LPDs contain EBV-DNA and express viral gene products such as EBV-encoded small RNAs (EBER), EBV plays a role in the pathogenesis of malignancies, including non-Hodgkin lymphoma (NHL) and Hodgkin lymphoma (HL) in PWH (11-13).
Material and methods:
- 6. Sample preparation: In centrifugation give rcf value instead of rpm
Author Response:
Comment accepted. We addressed the reviewer's comment on lines 132 (all changes identified as track changes)
- 9. DNA extraction: Give the sample amount in number of PBMC instead of μl of PBMC
Author Response:
Comment accepted. We addressed the reviewer's comment on lines 156 (all changes identified as track changes)
Discussion:
- Presence of EBV DNA in peripheral blood is a marker of reactivation, not co-infection. All EBV- seropositive PWH are co-infected.
Author Response:
Comment accepted. We addressed the reviewer's comment on lines 313 (all changes identified as track changes)
Conclusions:
- EBV load in PWH was significantly higher.
Author Response:
Comment accepted and addressed. All changes are identified as track changes.
Abstract:
- EBV is a risk factor for HL and NHL only in HIV- positive individuals. Rather than at higher risk of acquiring EBV infection PWH lose immune control of EBV latency.
Author Response:
We appreciate this comment and have addressed this with new text added to the Abstract section (see red/italicized text below and in the manuscript: Lines 23-24).
People with HIV infection (PWH) are at increased risk for EBV-associated malignancies such as HL and NHL. Nevertheless, there are limited data on the burden of EBV among this population group in Ethiopia.
Reviewer 3 Report
Epstein-Barr virus belongs to oncogenic viruses. The high risk group of EBV infection are people living with HIV. The authors assessed the prevalence of EBV infection among HIV-infected adults in Ethiopia. Two hundred sixty people were enrolled in this study, including 179 HIV-positive and 81 HIV-negative individuals. EBVCA anibodies were tested by ELISA. EBV-DNA levels were tested by quantitative real-time polymerase chain reaction (q-PCR) assays .The authors found that a higher HIV viral load in PWH was associated with an increased risk of EBV DNA positivity. This may be associated with a higher risk of developing EBV-related cancers. Minor errors should be corrected before the manuscript is accepted for publication.
Minor comments: Line 136 algorism - should be corrected - algorithm Line 157 syber Green - should be corrected SYBR Green Master Mix
the title of journals in the bibliography should be unified -References Abbreviated Journal Name
Author Response
Response to Reviewer # 1 Comments:
Minor comments:
- Line 136 algorism - should be corrected - algorithmLine 157 syber Green - should be corrected SYBR Green Master Mix
Author Response:
Comment accepted. We corrected the spelling and addressed the reviewer's comment on lines 147 and 156 (all changes identified as track changes)
- the title of journals in the bibliography should be unified -References Abbreviated Journal Name
Author Response:
We appreciate this point and have addressed this with edited text added to the Reference section (all changes identified as track changes).
